# An Attempt to Understand Stainless 316 Powders for Cold-Spray Deposition

**Neeraj S. Karmarkar *** **, Vikram V. Varadaraajan, Pravansu S. Mohanty and Sharan Kumar Nagendiran**

Department of Mechanical Engineering, University of Michigan-Dearborn, Dearborn, MI 48128, USA
* Correspondence: nkarmark@umich.edu; Tel.: +1-313-686-1888

**Abstract:** Cold gas dynamic spray (CS) is a unique technique for depositing material using high-strain-rate solid-state deformation. A major challenge for this technique is its dependence on the powder's properties, and another is the lack of standards for assessing them between lots and manufacturers. The motivation of this research was to understand the variability in powder atomization techniques for stainless steel powders and their subsequent properties for their corresponding impacts on CS. A drastic difference (~30%) was observed in the deposition efficiencies (DEs) of unaltered, spherical and similar sized stainless steel (316) powders produced using centrifugal (C.A) and traditional gas atomization (G.A) techniques. The study highlights more the differences on a precursor level. Using recent advancements in large scale statistical measurements, such as laser diffraction shape analysis and μCT scanning; and traditional methods, such as EBSD and nanoindentation, an attempt was made to understand the powder's properties. Insights on powder size and shape were documented. Significant differences were observed between C.A and G.A powders in terms of grain size, fraction of high-angle grain boundaries (HAGBs) and nanohardness. The outcomes of this study should be helpful for understanding the commercialization of the cold-spray process for bulk manufacturing of powder precursors.

**Keywords:** cold spray; 316 stainless steel powder; atomization; identical morphology; EBSD; μCT; nanoindentation



## 1. Introduction

Cold spray (CS) is a solid-state deposition process which is used to manufacture protective coatings or for repair and restoration and near net shape consolidation applications [1–3]. Known as the kinetic-energy-based additive manufacturing solution, the build-up occurs when powder particles impact and plastically deform the prior layer. This high-strain-rate deformation results in a combination of plastic deformation and hydrodynamic jetting [4] and adiabatic shear instability [5] on the surface of the powder, creating anchoring points that result in exceptionally high bond strength that the CS process is known for [6]. The coating also benefits from mechanical interlocking [7] and break up of its peripheral oxide layer [8], which enables metallurgical bonding [6,9]. The concept of critical velocity is widely accepted as a threshold value above which deposition occurs. This velocity is defined by transition from elastic particle rebound to a plastic deformation regime [10] and is dictated by its material properties and processing conditions. The applications of CS have expanded over 30 years; further strides are being conducted in-order to commercialize cold-spray as a powder consolidation technique to make geometric parts with desirable properties [11].

Widespread commercial adoption of this technique requires significant reductions in development and final manufacturing costs. Being a plastic-deformation-based manufacturing process, the starting microstructure of the powder plays a major role in the CS coating process. The microstructure, in turn, is dictated by the powder production or synthesis technique. Figure 1 shows a short list of powder characteristics/parameters affecting

the cold-spray deposition. Parameters such as size, morphology, microstructure, phase, porosity and oxide content have been systematically shown to affect the deformation process [12–15]. A thorough review of these factors was recently presented by Nastic et al. [16]. As mentioned by other researchers, extremely limited data have been reported on the effect of the starting microstructure on CS—the phase, grain size and porosity of the starting powder particles. Adding to the complexity is the variability in production of powders between manufacturers and within production lots. Brewer et al. [17] have studied these in detail and have raised the need for better standards for powder quality. Therefore, development of standards and tailoring techniques for the powder is critical for CS application. This will enable higher deposition efficiency (DE) and reduce manufacturing costs. There are challenges in the tailoring of feed stock powder. One is the costs associated with the tailoring techniques which usually require fluidization in inert gasses at high temperatures. The other is scalable quality control techniques to analyze the powders for the CS process [18–21].

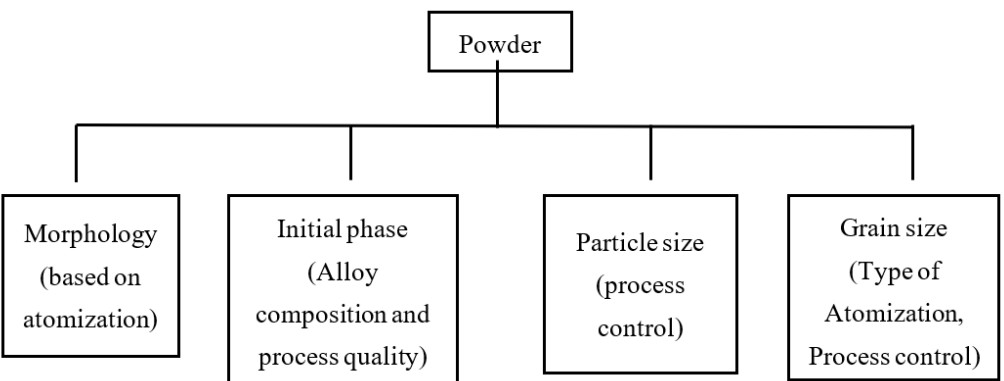

**Figure 1.** Parameters of powders affecting cold-spray quality (considered for this study).

Cold spraying of austenitic stainless steels is widely used to improve surface corrosion resistance and has been well studied by several researchers. Powder size [13], nozzle dimensions [22], mixed powders [23] and laser assisted [24] spray have been well documented. In a CS study using water atomized (W.A) and gas atomized (G.A) stainless steel powders, extensive deformation was observed for the W.A powder, leading to more hardness in its coating [25]. The irregular powder morphology results in better drag characteristics, resulting in higher particle velocities [25]. Brewer's study [17] compared three different G.A stainless steel 316 powders with different particle distributions and observed different initial phases in the powders. They recorded variation in atomized powders between batches/lots and manufacturers. These have been systematically identified as factors that affect CS deposition.

Centrifugal atomization (C.A) is a process that utilizes centrifugal forces to break up a rotating sheet of liquid into fine droplets. While C.A powders are widely used in additive manufacturing and powder metallurgy, limited information is available on their use with CS [26,27]. Unlike two-fluid atomization techniques, the centrifugal process can be controlled better and can achieve a narrow particle-size distribution (PSD). This process also results in higher solidification rates compared to $10^8$ $°Cs^{-1}$ for G.A, and $10^6$ $°Cs^{-1}$ to $10^7$ $°Cs^{-1}$ for W.A [28]. C.A typically produces powders with higher densities, less sphericity and less roundness compared to gas atomization. Hence, comparing the properties of C.A and G.A powders would improve our understanding of their impacts on CS and assist in the development of powder production for CS. This article emphasizes powder assessment, and significant coating properties are presented.

## 2. Materials and Methods

### 2.1. Cold-Spray Deposition

Commercially atomized powders from two reputed manufacturers were used in this study. To prevent readers from getting distracted from the results, the powder manufacturers are not disclosed. The sample names in the article will be further abbreviated as G.A (Gas Atomized 316 powder and coating) and C.A (Centrifugal Atomized 316 powder and coating).

The chemical compositions of the powders are listed in Table 1. To delineate the differences in deposition characteristics, coatings were sprayed on large aluminum 6061 plates ($12'' \times 12'' \times \frac{1}{2}''$) to minimize the influence of target heating. The CS deposition parameters were kept identical for both types of powders, as mentioned in Table 2a. The deposition efficiency of the process was simply calculated as the ratio of the coating weight vs. the powder amount fed for the respective cycle time, excluding technological losses. During deposition, the nozzle was stationery and the substrate was guided on a 2-axis CNC system (x—linear; y—rotational). A 50% overlap of the coating with respect to the nozzle width was maintained during deposition of both the powders for the subsequent linear and rotary motion of the substrate. The rotation speed was varied to maintain constant linear surface velocity of approximately 12 mm/s across the surface. The technological losses thus accounted for the amount of powder not deposited on the substrate during the linear and rotational motion using CNC. A convergent–divergent nozzle with a rectangular cross-section (expansion ratio 8.79) was used in this study. The details of the nozzle and deposition parameter optimization can be found in our previous study [24]. The nozzle was fabricated out of SS304. However, to avoid nozzle clogging due to high nickel content in SS316, the nozzle body was nitrided to provide a hard protective layer. Nitrogen was used as process and carrier gas. Deposition was conducted with 3.44 MPa process gas pressure and 0.47 MPa carrier gas pressure. Powder was fed using a Thermach AT-1200HP (Appleton, WI, USA) powder feeder. Substrates were degreased with alcohol; no additional surface preparation was performed. Coatings were deposited to a thickness of 1.3 mm. C.A powder required additional deposition time to achieve the same thickness as the G.A powder (Table 2b).

**Table 1.** Chemical composition of powders used for cold spray.

| Composition | Fe | Cr | Ni | Mn | C | Mo | Si | P | O | N |
|---|---|---|---|---|---|---|---|---|---|---|
| G.A powder | Bal | 16.9 | 11.59 | 0.46 | 0.02 | 2.39 | 0.7 | 0.04 | 0.03 | 0.01 |
| C.A powder | Bal | 17 | 11.5 | 1.35 | 0.03 | 2.16 | 0.87 | 0.037 | 0.04 | 0.0588 |

**Table 2.** Deposition parameters and experimental results.

| (a) | Temperature (°C) | Pressure (MPa/psi) | Standoff (mm) | Carrier Gas (MPa/psi) | Powder Flow(g/min) |
|---|---|---|---|---|---|
|  | 600 | 3.447/500 | 10 | 0.47/68 | 20 |

| (b) | Powder | Layers to achieve ~1.3 mm coating thickness | Deposition time | Deposition Efficiency (%) |
|---|---|---|---|---|
|  | G.A | 4 | 32 min 28 s | 70.26 ± 0.3 |
|  | C.A | 6 | 48 min 52 s | 41.33 ± 0.92 |

### 2.2. Characterization

Microstructural characterization of the powders and cold-sprayed coatings was performed with scanning electron microscopy (secondary/back-scattered images) on a Hitachi S-3600N SEM to analyze the grain size and the quality of the powders and the coating. Standard metallographic procedures were followed using an epoxy resin to prepare cold mount samples for the SEM. The internal structure of each powder was evaluated by mounting, polishing and etching. A swab etch was performed on the cross-section of the powder using diluted $HNO_3$ (1:9) as the etchant with 10 s of etching time. Etching was carried out to reveal the microstructure grains of the precursor (powder). A Rigaku

Miniflex X-ray diffractometer (Cu Kα radiation λ~1.5402Å) was used to determine the phases in the powders and the coatings. The Scherrer equation was used to calculate the average crystallite size for all peaks with K = 0.9 (Scherrer constant).

A Tescan Mira SEM coupled with an EBSD detector with an EDAX Hikari camera was used for microstructure measurements. An accelerating voltage signal of 30 kV was used with a high beam intensity. Diffraction patterns were indexed using crystallographic information imported from the ICSD database [29]. The crystallographic information is mentioned in Table 3. Data were collected with step sizes of 0.04 μm (fine scans) and 0.1 μm (preliminary scans) at a working distance of 15–17 mm. Individual particles were examined over an area of ~4000 μm². The EBSD scans were later processed using TSL OIM analysis, and only the patterns with C.I (Confidence Index) > 0.2 were mapped.

**Table 3.** Crystallographic information for phases used for EBSD indexing.

| Phase | a (Å) | b (Å) | c (Å) | α | β | γ | Crystal System | Space Group | ICSD # |
|-------|-------|-------|-------|-----|-----|-----|----------------|-------------|--------|
| Austenite | 3.66 | 3.66 | 3.66 | 90° | 90° | 90° | Cubic | Fm-3 m (225) | 631733 |
| Ferrite | 2.87 | 2.87 | 2.87 | 90° | 90° | 90° | Cubic | Im-3 m (229) | 180969 |

A Microtrac Sync system was used to measure the PSD (particle-size distribution) and shape analysis using laser diffraction (LD) and dynamic image analysis, respectively (Microtrac Retsch GmbH, Duesseldorf, Germany). A ZEISS Versa 620 X-ray microscope was used to characterize the powder in 3D. The scan used a voltage of 140 kV and power of 21 W to image the powder prepared in glass capillary tubes. A pixel size of 0.75 μm was obtained with the 4× objective. The powder was segmented using commercially available software from Zeiss—Powder Analyzer. Visualization was completed in TXM3DViewer, allowing for 2D cross-sections and 3D views to be obtained.

The micro hardness was measured using Vickers Hardness equipment (Streurs, Westlake, OH, USA). Loads of 10 gF and 300 gF for 20 s were applied for hardness measurement on powder particles and on the cold-sprayed coatings, respectively. The indents on the powder's cross-section were measured on the same particle and different particles to ensure homogeneity of the precursor micro-hardness. Micro-hardness on coatings was determined at 3 different locations, viz., 0, 450 and 900 μm away from the substrate. An average of 5 readings at each offset from the substrate has been reported.

Mechanical properties of the powder precursors were also determined using a nanoindentation technique. A hysitron 950 tribometer equipped with a standard Berkovich probe (half-angle 65.23° $E_{indenter}$ = 1140 GPa, $\nu_{indenter}$ = 0.07) was used for measurements. Only larger particles in the range of 35–50 μm were considered for nanomeasurements to avoid the effect of immersion in epoxy [30]. Indentations were performed using a constant displacement of 250 nm to avoid any surface effects. The applied load was held for 5 s. Multiple indents spaced at significant distances were made in same particles to prevent local strain effects during indentation.

The tribological properties of the coatings were measured using a commercial reciprocating ball-on-disc setup (CSM Tribometer, Switzerland) according to ASTM G99. Machined coatings were used for the testing in order avoid excess loading on the tribometer. A 6 mm tungsten carbide ball (McMaster Carr) was used as a friction medium. The test was conducted at room temperature (~24 °C) with a 3.25 cm/s sliding speed and a load of 9 N for 25,000 cycles. Corresponding wear profiles were later measured for a travel span of 6 mm using Mitutoyo Surf test equipment. The resulting plot of Ra value vs. travel span length is reported for repetitive and uniform measurements.

Adhesion strength on the coating was measured by a hydraulic tensile adhesion tester. The equipment used was P.A.T (precision adhesion testing) by DFD instruments, UK. The test was performed as per ASTM C633 [31]. An FM-1000 epoxy adhesive film (Cytec Industries, Woodland Park, NJ, USA) was used. Grit blasting was performed on the contact face of the bond pin. The film was glued to the coating at 175 °C for 2 h (later air

cooled at ~25 °C), followed by the pull test. A total of 4 measurements were taken from each of the coatings. It was ensured that the test results represent the values at separation of the coatings from their respective substrates; other results with partial delamination were discarded.

## 3. Results

### 3.1. Powder Characterization

The feedstock powders obtained from both C.A and G.A techniques were spherical. This is shown in the SEM micrographs of as-received and etched powder particle cross sections in Figure 2. The G.A powder showed many agglomerated satellites (Figure 2a). The outer surface structure of G.A powder did not reveal complete structure (Figure 2b). The cross-section (Figure 2c) shows that G.A powders had a near complete transition from the columnar dendrites to multi-crystalline anisotropic grains due to the cooling time during atomization, ranging from 5 to 10 μm. The C.A powder showed less agglomerates (Figure 2d). The outer surface structure of C.A powder (shown in Figure 2e) showed a cellular dendritic combination [32,33]. The cross-section shows the majority of the internal features of the C.A powder to have been under 5 μm (shown in Figure 2f). The differences in the solidification rates during atomization led to different dendritic networks in the powders. Similar features were present in both G.A and C.A powders—e.g., the z-contrast present along the grain boundaries [32–35]. This was further assessed using EBSD analysis, as presented later in this study.

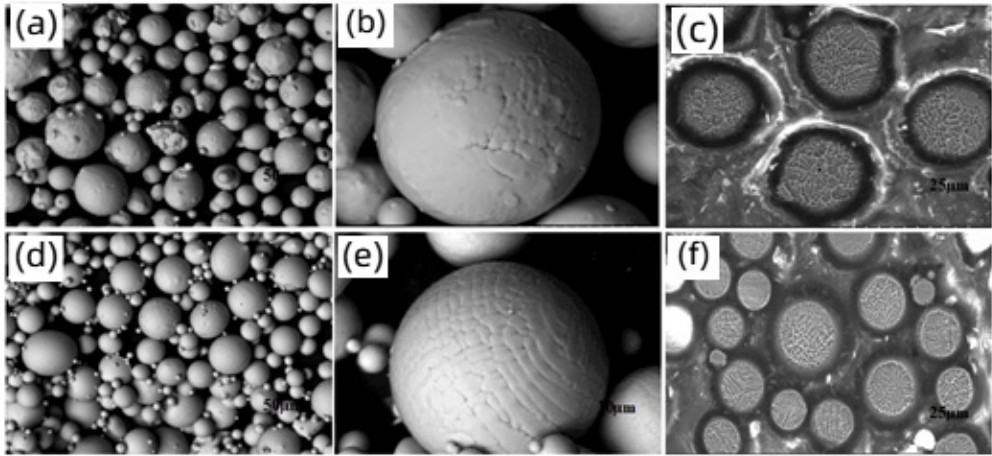

**Figure 2.** BSE image and etched crossection at 500×, 2500× and 1000× magnification of G.A powder (**a**–**c**) and C.A powder (**d**–**f**).

PSDs collected using LD and μCT are shown in Figure 3. The powder flow properties obtained from manufacturer data sheets are summarized in Table 4. The difference between the two measurement techniques is that they compute the particle diameter from the actual measurement, which is the area (LD), and volume using 2D rotation (μCT), respectively. For the C.A powder, the PSDs calculated using particle area and volume are similar. For the G.A powder, there was a significant difference in the diameters calculated from scanned volume vs. scanned area. This difference highlights the limitation of optical techniques and image analysis, which are limited to two dimensions. Small differences between G.A and C.A measurements could have been primarily due to the agglomeration observed in G.A. These measurements differ slightly from the data provided by the manufacturer. This highlights the limitations of each PSD technique that is available in the industry.

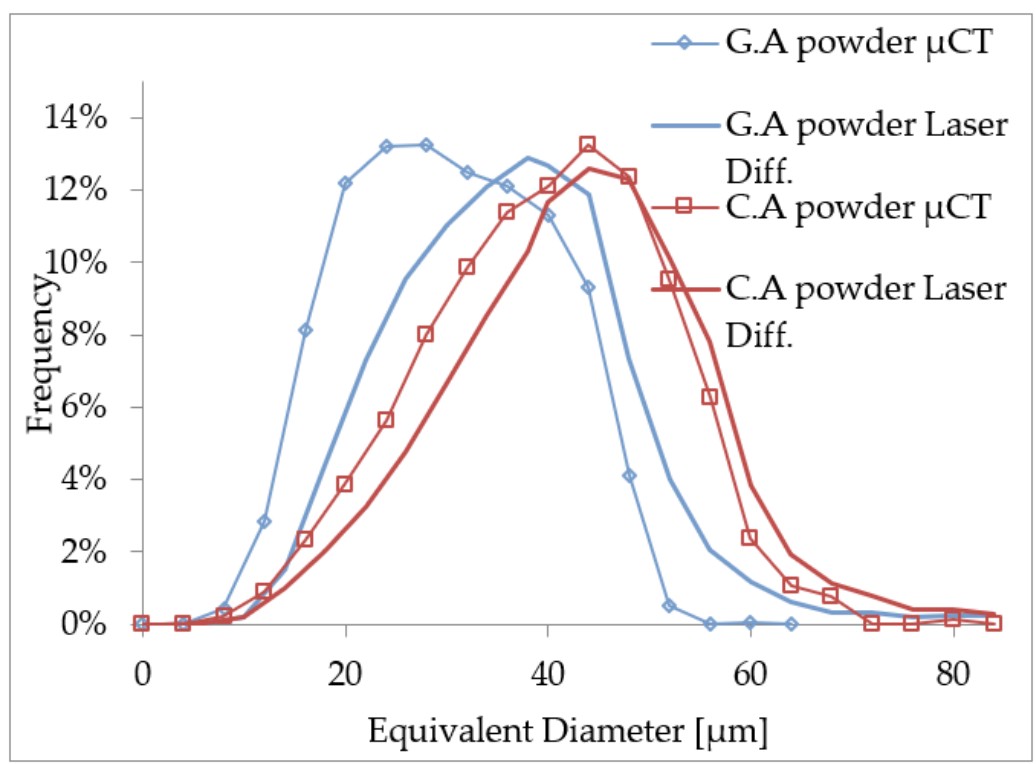

**Figure 3.** Particle-size distributions for both stainless steel powders obtained using laser diffraction and μCT techniques.

**Table 4.** Physical properties of the powders used in this study (manufacturer's specifications).

| Powder | D10 (μm) | D50 (μm) | D90 (μm) | Hall Flow (s) | Apparent Density (g/cm$^3$) |
|---|---|---|---|---|---|
| G.A | 20 | 31 | 46.9 | 16 | 4.10 |
| C.A | 18.4 | 30.4 | 48.4 | 13 | 4.29 |

A snapshot of the morphology of each powder, along with sphericity and roundness characteristics collected using LD, is shown in Figure 4. These snapshots were taken from a camera attached to Microtrac Sync equipment. The statistics were calculated from well over 50,000 particles. Non-spherical atomization artifacts in G.A and C.A powders are shown in Figure 4a,b but account for a statistically small fraction. Figure 4c,d show the roundness and sphericity of both powders. Sphericity is the ratio of minimum circumscribed and maximum inscribed circles and for the roundness is the ratio of area of the particle to the area of a circle with the same convex perimeter.

While typically, the centrifugal powders exhibited higher sphericity and roundness, when segregated based on calculated particle diameters, both powders were identical below 35 μm. Above this size, the centrifugal powder was more spherical and exhibited higher roundness. These details are shown in Figure 5. Further insight into particle's internal porosity was obtained from a μCT scan and is shown in Figure 6. A snapshot of the scanned volume is shown in Figure 6a,b. The individual pore volumes were calculated using the Zeiss software and are presented in Figure 6c. The C.A powder had a significantly lower fraction of fine pores with size of 15 microns or less and a larger fraction of coarse pores compared to gas atomized powder. Correlation of the particle volume to the pore volume was not available. However, given that particle diameter must be greater than pore diameter, at dimensions of 30–60 μm, the C.A powders exhibited slightly higher porosity compared to gas atomized poweders.

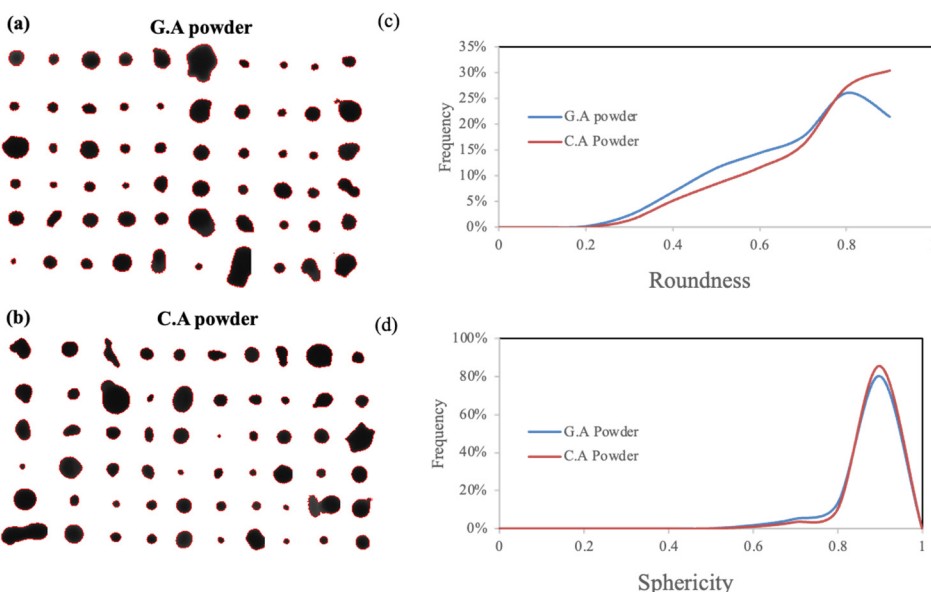

**Figure 4.** Snapshot of the powder particles from the shape analyzer: (**a**,**b**) roundness, (**c**) sphericity and (**d**) distributions of both powders.

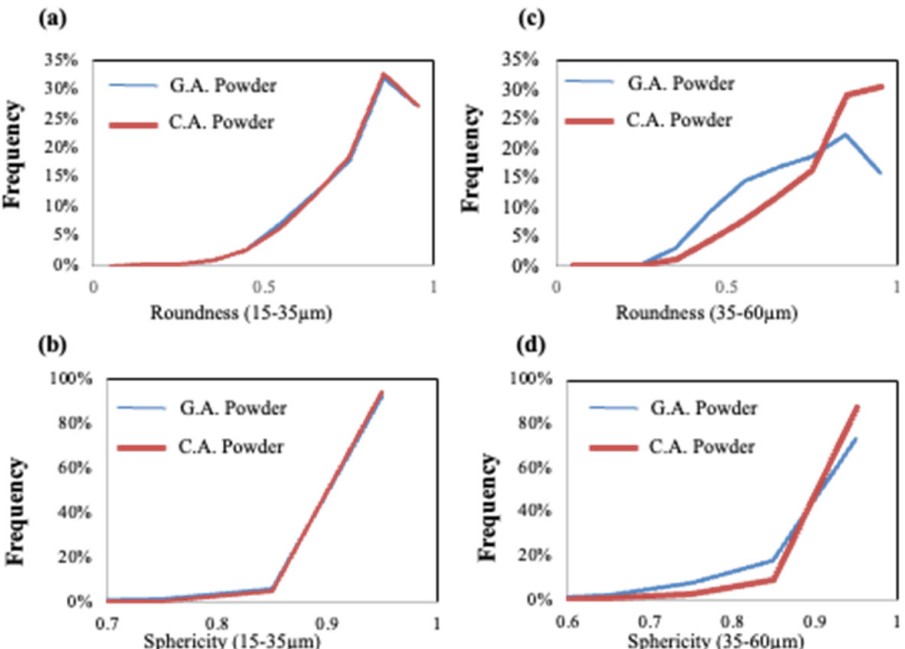

**Figure 5.** Roundness and sphericity distributions of the particles analyzed between 15–35 μm (**a**,**b**) and 35–60 μm (**c**,**d**).

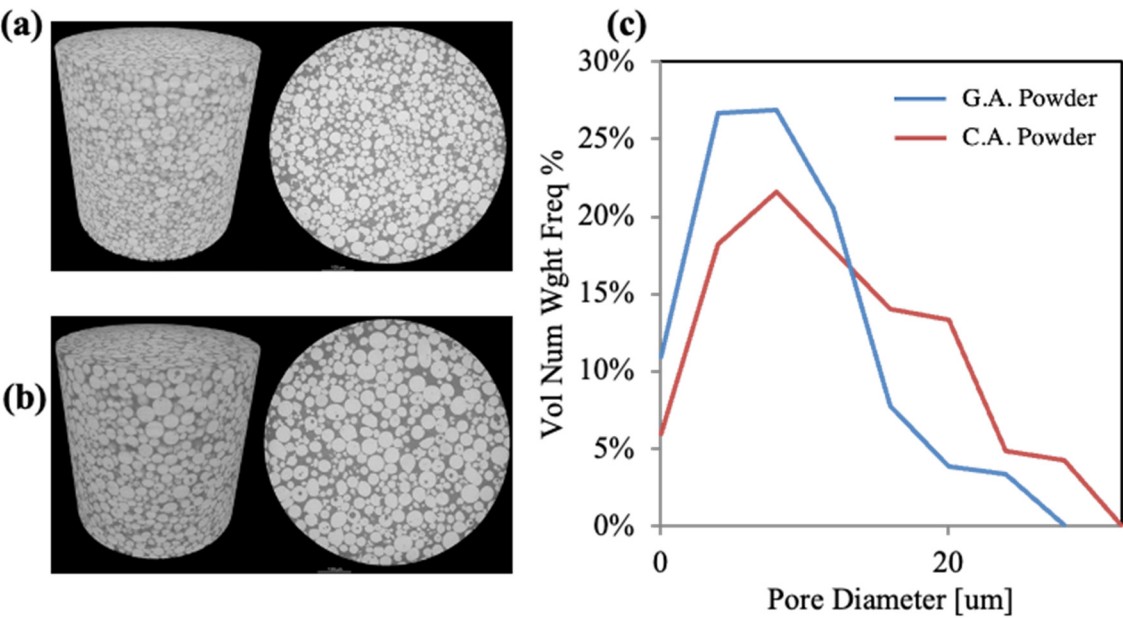

**Figure 6.** μCT scan results—snapshot of the scan volume: (**a**) G.A powder and (**b**) C.A powder. (**c**) Pore size distribution.

Powder X-ray diffraction (Figure 7) indicated minor amounts of ferrite and austenite in the as-received stainless steel powders (G.A and C.A). From the diffraction pattern, the fraction of residual ferrite was higher in the C.A powder. This could have been due to higher solidification rates that can be achieved in centrifugal atomization. Brewer et al. [17] have analyzed phase content within gas atomized stainless steel powders and found that powders with overall smaller size distributions of particles had high fractions of the ferritic phase due to their high solidification rates.

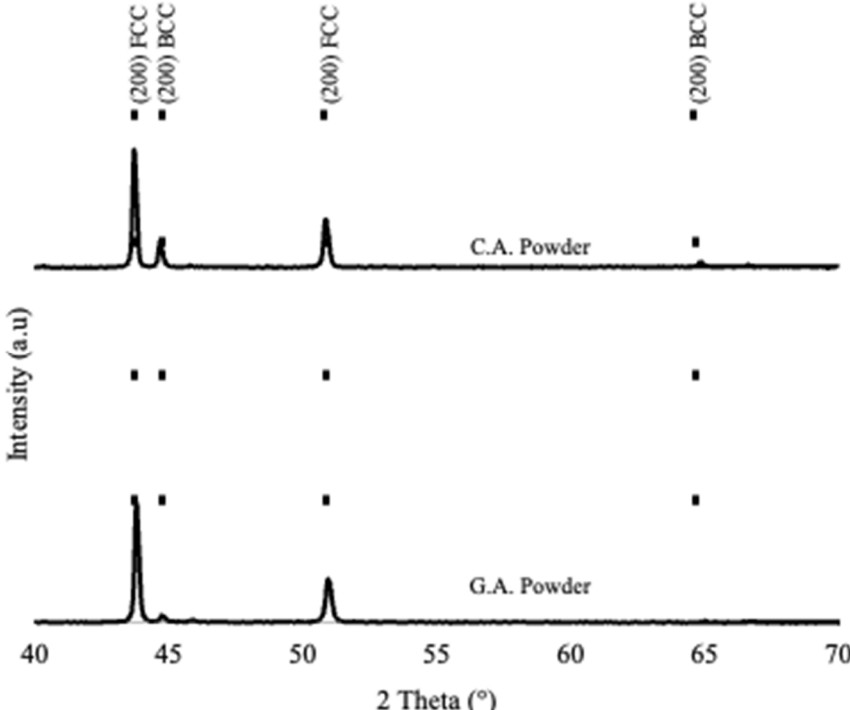

**Figure 7.** XRD spectrum of as-received powders.

The EBSD characterization of the powders in shown in Figure 8. About 10–15 particles were analyzed from each of the atomization techniques. Figure 8a,d show typical IPF maps obtained from each of the powders. Corresponding phase and IQ (image quality) maps are shown in Figure 8b,c and Figure 8e,f, respectively. The phase maps clearly show that the ferrite content was present in smaller fragments of the powder and can be associated with higher solidification rates of the particle. The grain boundary angle is superimposed on the IQ map. The low-angle grain boundaries (LAGB) < 15 deg and high-angle grain boundaries (HAGBs) are shown in blue/green and red, respectively.

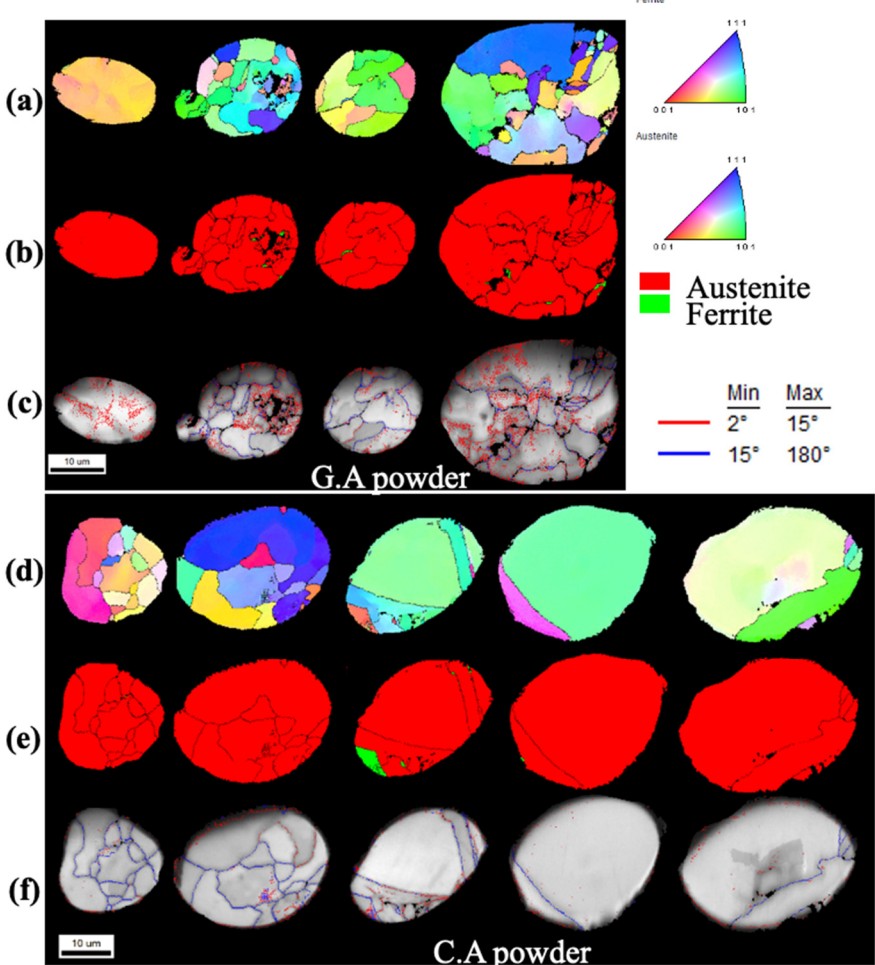

**Figure 8.** EBSD data IPF (**a**,**d**); phase map (**b**,**e**); IQ map with red (>2°, <15°) LAGBs and blue HAGBs (>15°) (**c**,**f**) for G.A and C.A powders, respectively.

The mis-orientation angles and grain size distributions of different powders were extracted using OIM software and plotted in Figure 9a,b. The effect on grain structure due to differences in solidification kinetics achieved in gas and centrifugal techniques can be observed in terms of grain boundary angle and size distribution. From the obtained data set (Figure 9a), we observed significantly lower fractions of grain boundaries < 5 deg or sub-grain boundaries in the centrifugally atomized powder. The absence of low-angle grain boundaries could have been due to higher solidification rates. A similar conclusion has been derived by other researchers [36]. A similar effect was found in the grain size distribution (Figure 9b). The atomization techniques produced grains of a similar size, but a lower solidification rate for the G.A powder appears to have resulted in a high fraction of LAGB, leading to grain coarsening. While both powders had grain sizes between 1 and 12 μm, larger grains were present in the C.A powder and can been seen in Figure 8d–f. The

C.A microstructure had dendrites, which led to larger grains relative to the equiaxed grains in the G.A powder.

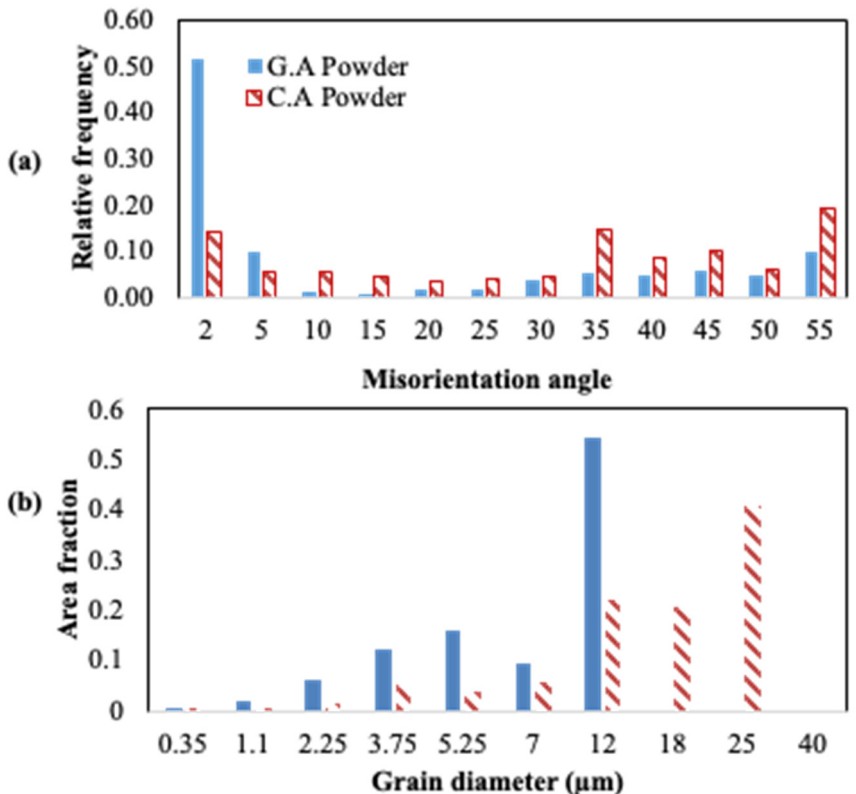

**Figure 9.** Total misorientation (**a**) and grain size distribution (**b**) obtained from the G.A and C.A powder EBSD scans.

Table 5 shows the crystallite sizes of G.A and C.A powders and their respective coatings. Table 6 represents the micro- and nanohardness measurements of both powders. These hardness values were within a range for austenitic stainless steels reported in literature [17,34,37–41]. Nanohardness was measured on the cross-section surfaces of both powder particles. Various areas within the particle were indented to delineate the differences within the particle. The nanoindentation curves are shown in Figure 10. This curve represents an average and $3\sigma$ distribution in the loading section of the test. The C.A particles required a higher load for the given indenter displacement. Average hardnesses of 2.65 and 2.87 GPa were recorded for G.A and C.A powders, respectively. This higher hardness in the C.A powder could have been due to the higher fraction of HAGBs observed with EBSD measurements. The difference in cooling rates during atomization could have been another contributing factor. Other differences in hardness could have been due to grain orientation with respect to nanoindents, which has been studied by Roa et al. [42].

**Table 5.** Crystallite sizes of both powders and coatings calculated from the XRD data.

| | Crystallite Size (nm) Powder | Crystallite Size (nm) Coating |
|---|---|---|
| G.A | 50.39 | 16.36 |
| C.A | 61.78 | 18.30 |

**Table 6.** Micro- and nanohardness measurement data on the powders.

| Powder | Micro Hardness (Vickers 10 gF) | Nano Hardness (GPa) | Stiffness (μN/nm) |
|---|---|---|---|
| G.A | 273.14 ± 23.39 | 2.65 ± 0.12 | 215.9 ± 16.1 |
| C.A | 279.4 ± 14.54 | 2.87 ± 0.11 | 248.88 ± 14.08 |

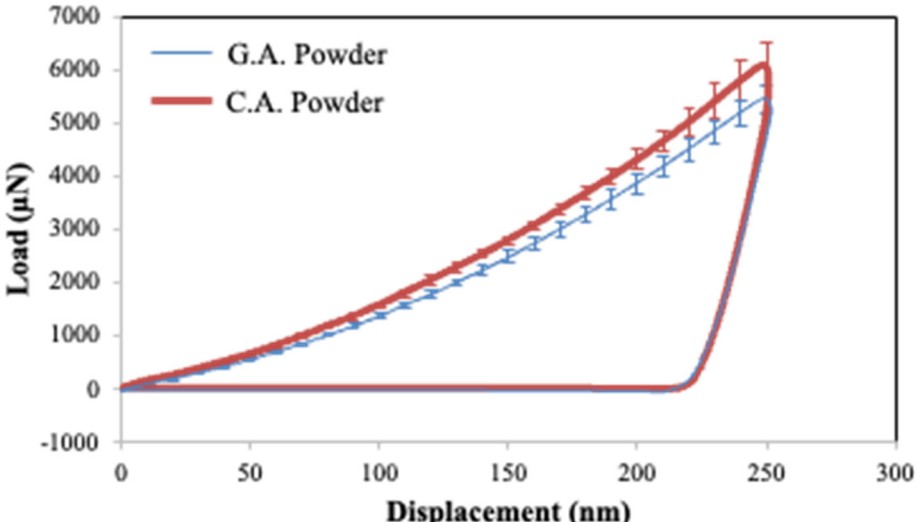

**Figure 10.** Load-displacement response of nanoindentation test on C.A and G.A powders.

### 3.2. Coating Characterization

The DE of C.A powders was significantly lower compared to G.A powders, and we required additional deposition layers to achieve the 1.3 mm coating thickness. Cross-section images of both the coatings are presented in Figure 11. The arrow in each image indicates the direction of particle travel during deposition. Inter-splat voids and boundaries were present in the coatings. It is typical in CS for a softer substrate to extrude around a harder particle [43]. Since a soft substrate (aluminum) was used, the initial layer of particle–substrate bonding occurred due to a phenomenon termed surface-scrubbing. This results in extrusion of aluminum in between the particles, which limits the hardness of the bottom layer, as marked in red in the figure [44]. A similar effect was present for the G.A coating. In comparison, its magnitude and occurrence were limited in the C.A coating.

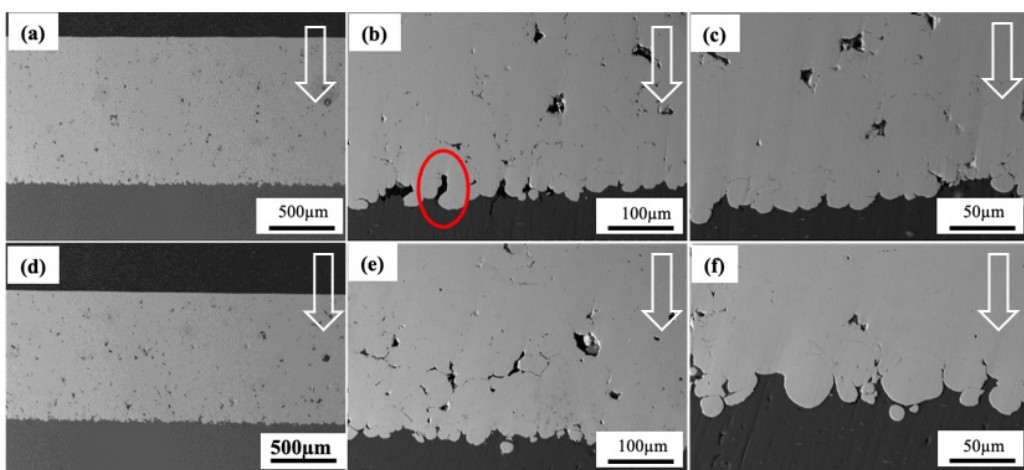

**Figure 11.** (**a**–**c**) BSE images for GA coating; (**d**–**f**) BSE images for CA316 coating.

Figure 12 shows the XRD of the respective coatings. A complete austenite phase was observed in both coatings. Peak shift and broadening are frequently observed in cold-sprayed coatings [45]. The shift occurs due to intensity of the residual stress, which increases proportionately to the degree of cold work. The peak shift of ~0.8° is visible in the G.A coating and reveals a higher degree of residual stress accumulated in the coating compared to the C.A coating. Significant peak broadening was observed in the FCC (γ) phases of both the coatings compared to the peaks of the precursors. It led to a change in the crystallite size compared to the precursor. A lattice parameter of ~0.41–0.42 nm was calculated from the XRD patterns of both the powders and their respective coatings. Crystallite size reduced significantly from the precursor to its respective coating, as mentioned in Table 5. The C.A powder had a greater crystallite size than G.A. The difference was ~10 nm, which reduced to ~2 nm in the coatings. The broadening and reduction in crystallite size for a coating corresponds to a non-uniform micro-strain [46,47], which is a typical phenomenon observed in CS [48].

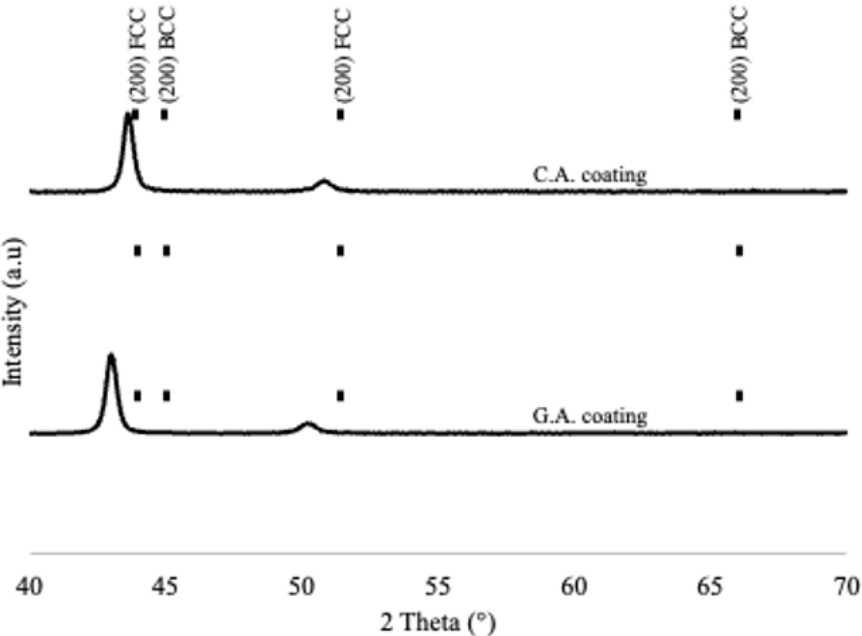

**Figure 12.** XRD spectra of coatings.

Microhardness measurements for powders and the respective coatings are listed in Table 6 and Figure 13. Hardness values of the G.A and C.A coatings were from ~217 to 260 HV and 219 to 220 HV, respectively. These values are well within the typical range for cold-sprayed stainless steel coatings [13,17]. The ultimate tensile strength (UTS) was derived from Vickers hardness measurements [49] and was calculated to be 695–710 MPa for C.A and from 695–895 MPa for G.A coatings. As is usually observed for cold-sprayed coatings on softer substrates at the bond line, the coating's hardness was relatively low, which then increased with coating thickness, as the deposition mechanism switched from impingement to plastic deformation [50]. Both C.A and G.A coatings showed similar low hardness levels near the bond line. However, with increasing the distance from the bond line, only the G.A coating demonstrated a corresponding increase in hardness. The C.A coating had a monotonic hardness value of ~220 HV from the substrate to the free surface, indicating a lack of cold work and a low degree of deformation.

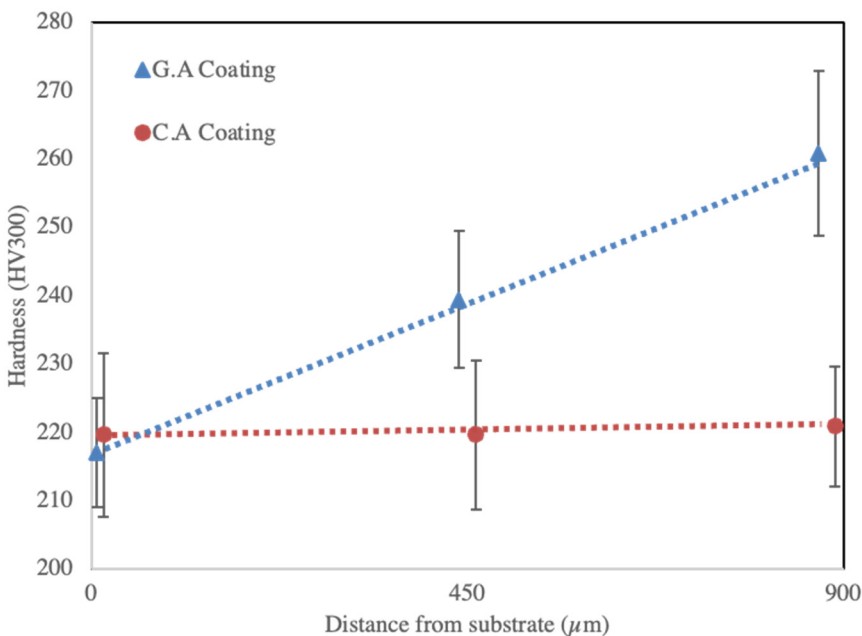

**Figure 13.** Micro-hardness values of coatings.

Table 7 represents results of the bond test. The G.A coating showed higher bond strength than the C.A coating. The location of bond failure is critical to understanding the bond quality of a coating. The G.A coating experienced adhesive failure of the FM1000 epoxy. For the C.A coating, cohesive failure occurred until the initial layers of deposition. This shows that the bond strength for G.A coatings could be greater than 70 MPa, whereas C.A coatings failed to prove 45 MPa of tensile strength. The lack of deformation or cold work in the C.A coating observed from the monotonic hardness levels shows that poor mechanical anchoring reduced the bond strength of the C.A coating significantly.

**Table 7.** Bond strengths of the coatings.

|  | Bond Strength (MPa) |
|---|---|
| G.A Coating | $68.125 \pm 1.10$ |
| C.A Coating | $47.5 \pm 0.76$ |

Figure 14a shows the plot of the coefficient of friction across laps/cycles. The static coefficients of friction (COFs) achieved for both the coatings were between 0.7 and 0.9. The C.A coating exhibited relatively lower COF values compared to the G.A coating. Figure 14b represents the cross-sectional profile of the wear track (one profilometer for of each coating). The depth and width of the wear track of the C.A coating was wider and deeper than that of the G.A coating. The wear depths were 100 and 75 μm, respectively. A fast wear rate in a cold-sprayed coating is typically caused by poor bonding between particles, causing them to flake and increase surface roughness. Wear would also occur when the sliding 316 stainless steel ball interacts with the coating's pores, causing a shoveling effect, resulting in a higher wear rate and an increase in the COF. The COF depends on many factors, such as the mechanical properties of material, the surface roughness, the mutual dissolution of materials, the contact time, the lubricant film's properties and the elasticity of the tribo-system. Given identical test conditions and identical material composition for the coatings, similar COF values indicate that both coatings had similar surface roughness for the duration of the test. This shows that a more probable source for the higher wear rate of the C.A coating is its lower hardness levels. During the wear test, a softer coating

would result in a combination of material removal and compaction, causing minimal effect on surface roughness [51].

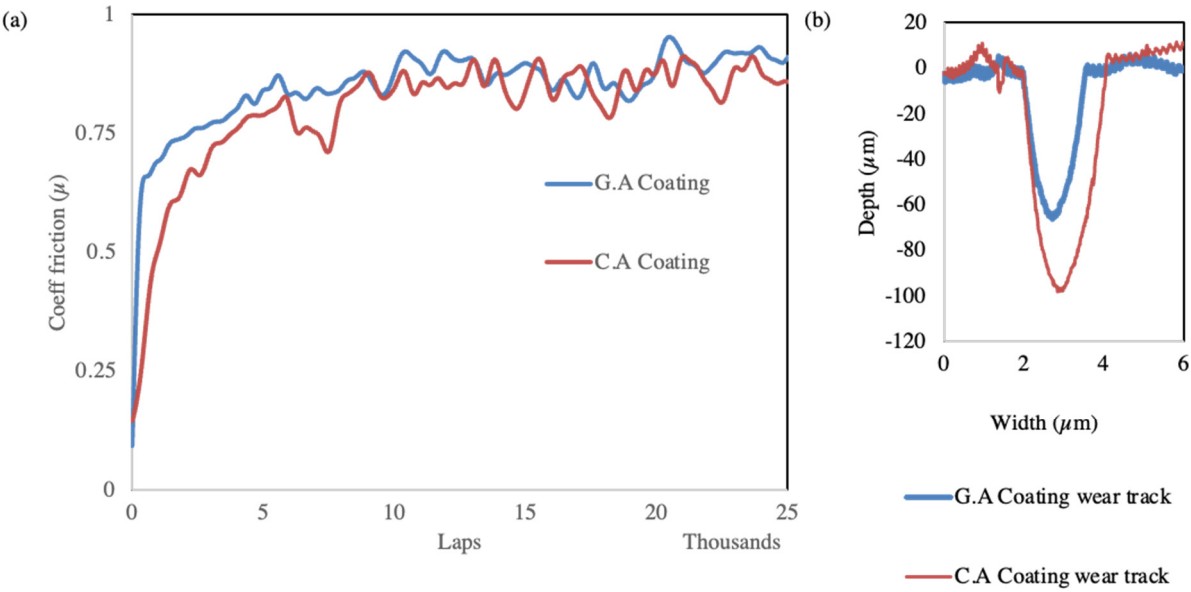

**Figure 14.** (**a**) Friction coefficients from G.A and C.A coatings over 25,000 laps; (**b**) measured wear track profile from after testing.

## 4. Discussion

Given that G.A and C.A yield particles of similar sizes and shapes, the initial observation of a drastic difference in deposition efficiency (DE) between them spurred the interest for this study. Identically sized "on the shelf" cuts of the powders were obtained with the intention of isolating the effect of the internal structure of the powder on DE. The discrepancy is the particle diameters measured using LD and μCT later revealed that the G.A powder is smaller than the C.A powder. The effect of powder size on DE was well documented by the authors in their previous research [52]. In general terms, it is well known in the cold-spray community that a finer particle distribution is not favorable for cold spraying. Ideally a distribution of 15–60 μm with D90 > 40 μm is recommended by the authors. The study was nevertheless continued in spirit of exploration to serve as a reference to the CS community.

Under identical deposition conditions, the C.A powder resulted in lower DE compared to coatings produced from G.A powder. The microhardness, bond, wear and corrosion studies demonstrated that the extent of cold work and the extent of plastic deformation were inadequate in the C.A coating. This suggests that the C.A powder requires a higher particle velocity to produce coatings comparable to those using the G.A powder. An estimate of the particle velocities will be provided in further studies.

Centrifugal atomization is known [53] for producing powders with both high densities and high solidification rates. A μCT scan enabled the measurement of porosity of the powder and has not been reported in prior works. The scan showed that C.A and G.A powders' porosities were similar. The effect of solidification rate was assessed with EBSD and nanohardness measurements. EBSD analysis carried out on the powder showed differences in grain size distribution and misorientation angles. About 50 powder particles were scanned, and a considerable fraction of the C.A powder particles were bicrystalline. As a result, these powders had larger grain diameters, as shown in the Figure 8. Using QCGD simulations, Suresh et al. [54] showed that jetting and material ejection and nucleation of defects were enhanced by the presence of polycrystalline materials with fine grain boundaries. Single crystal, and by that logic, bi-crystal material with a lower fraction of

grain boundaries would make it harder for defects to nucleate. Past research [52] performed by the current authors has also observed a reduction in cold-sprayed deposition ability with an increase in grain size during annealing of stainless steel 316 powders.

With respect to misorientation angles, the C.A powder had a much higher fraction of HAGBs This could have been due to the higher solidification rate, and has been reported in literature [36]. At a strain rate of 30,000 s$^{-1}$, which is low compared to the extreme strain rates experienced in CS deposition, Guha et al. [55] have shown through CPFEM that for fcc materials, HAGBs result in increased thermal softening, leading to zones of localized plastic strain. For shock loading applications, Nguyen et al. [56], using mesoscale models, have assessed the effect of misorientation angle on deformation and have concluded that misorientation angle alone is not sufficient descriptor. By decomposing the angle into tilt and twist grain boundaries, they found that tilt angles are dominantly related to the apparent strength of a GB. The effect of CSL boundaries on CS ability has also been recently discussed [57].

Nano-indentation measurements showed higher hardness value and stiffness for the C.A powder corresponding to its grain structure. Nanohardness, which reflects dislocation mobility, has been associated with CS deposit ability and DE (Ref. [16]). The higher fraction of HAGB in the C.A 316 powder could limit the dislocation mobility, resulting in higher nanohardness. Thus, this increase in resistance could explain the lower DE of the C.A powder compared to the G.A powder. The higher stiffness value of the C.A powder could also be correlated with the HAGB fraction shown in Figures 8c,f and 9a giving rise to the different deposition behavior.

## 5. Conclusions

The atomization techniques had a significant impact on the deforming mechanisms for a cold spray. Despite similar morphological characteristics, C.A powders had significantly less DE compared to G.A powders. Analysis of the C.A coatings revealed a lack of bonding between C.A particles, as was evident from its lower bond strength. Aside from the slightly larger powder size difference (which was identified much later in the investigation), EBSD and nanoindentation measurements revealed some distinct differences between the powders. This study found powder nanoindentation to be a reliable technique for evaluation of the cold-spray-ability of the material in terms of its deformability. Further evaluation of the EBSD data into tilt, twist and CSL boundaries is under review and will be reported in future studies. Further development in powder shape determination techniques using μCT could yield more insight into powder properties for cold spray. Low-CS-deposition-efficiency C.A powders require further validation in terms of other materials, manufacturers and production lots. A detailed microstructural evaluation of the cold-sprayed coatings would provide further supplementary information.

The obtained results can be connected to the variability of the production route of the powders. The microstructures of the C.A powders correspond to the higher cooling/solidification rates used during manufacturing. These in turn impacted their mechanical properties, internal porosity values, HAGBs, etc. Having powder atomization process parameters in hand (which are proprietary to the manufacturer) could go a long way towards understanding and optimization of powders for cold spray. This study thus provides comprehensive information for the powder precursor assessment using state-of-the-art characterization techniques. Such assessments can help us better understand the impacts on their respective coated materials for techniques such as cold spray. This will also help analyze the commercial feasibility of cold spray using all available production routes for powder manufacturing.

**Author Contributions:** All the authors contributed to the conceptualization, data, writing and revision. All authors have read and agreed to the published version of the manuscript.

**Funding:** This research received no external funding.

**Institutional Review Board Statement:** Not applicable.

**Informed Consent Statement:** Not applicable.

**Data Availability Statement:** Not applicable.

**Acknowledgments:** This research did not receive any specific grant from funding agencies in the public, commercial, or not-for-profit sectors.We acknowledge the University of Michigan College of Engineering for financial support and the Michigan Center for Materials Characterization (MC)2 for use of the instruments and staff assistance. We thank the folks from Microtrac MRB, Bernie Lucansky and James Pastore, for assisting with the SYNC laser diffraction measurements. We thank the management at Somnio Global LLC, Novi Michigan, for providing the resources to carry out the deposition, and William Walker for assistance with metallographic preparation.

**Conflicts of Interest:** This research did not receive any specific grant from funding agencies in the public, commercial, or not-for-profit sectors. On behalf of all authors, the corresponding author states that there is no conflict of interest for any aspect of the manuscript.

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
