# Peer review of "An Attempt to Understand Stainless 316 Powders for Cold-Spray Deposition"

_2674-0516, doi:10.3390/powders2010011_

Round 1
Reviewer 1 Report
Overall, the paper is good. It provides sufficient experiment analysis and data. It also has a good structure and findings. The paper corresponds well to the journal scope. I recommend publishing this research in Powders after some minor corrections as suggested below:
1. Line 25: Suggest to number the sections as this manuscript contains many sections and subsections. Can check if there is numbering in the journal template.
2. Line 39: typing error on powder.
3. Line 60: The lines of the tree diagram does not connect correctly, please check and revise.
4. Line 78: perhaps it is better to spell out w.r.t
5. Line 91: It may not be necessary to bold in-text table or figure. It is distracting. Please check the journal template if they also do so.
6. Line 93: typing error on the plate dimension.
7. Line 111: the caption of Table 2 is wrong. It is exactly same with Table 1.
8. Line 112: please reformat the table nicely.
9. Line 115: there is no need to capitalize S and B in secondary and backscattered image.
10. Line 119: the font seem to be different here, please check. And perhaps spell out dil.
11. Line 129: why the EBSD scanning were carried out with different step size? Need to describe when or for what are those 0.04 and 0.1 μm step size were used.
12. Line 131: need to describe or spell out C.I.
13. Line 134: was the long form of PSD mentioned previously? Please check and make sure that all the acronyms in the manuscript have been described or introduced before they are used.
14. Line 147: please give the number (can be estimated) in term of μm about the three offset locations from the substrate for microhardness measurement.
15. Line 197: no scale bar, the scale bar text is hardly visible. Please revise.
16. Line 204-207: Any reasons on why GA and CA case differs. Could it be due to the agglomeration of GA (line186)? Please discuss your thought on why the GA and CA case differs. And on which PSD method is more accurate?
17. Line 211: please also discuss on why the PSD result measured in LD and μCT slightly differ from the powder size statistic from manufacturer?
18. Line 220: for the graph, please draw the two lines using different thickness or line type. Graph in Figure 5 and 6 too.
19. Line 235: Figure 5 need graph legend
20. Line 238: in the μCT scan results, what are the white circle things? Are they grain size? Perhaps give a bit description in line 228.
21. Line 248: put GA first then CA, since this is the ordering all along. For the coating XRD result, please split the result to another figure and put it under coating characterization.
22. Line 250-251: typing error on 10-15 and were analyzed.
23. Line 252: need to describe or spell out IQ
24. Line 258: please draw both powders data using same scale bar length. So that the powder sizes can be directly compared.
25. Line 272: please give the thought on why larger grains were presented in the CA powder. Usually a faster cooling rate will result in smaller grain size. So, this result is interesting therefore thought or discussion on it is important.
26. Line 274: figure 9(b), need to change the blue color to lighter tone similar to figure 9(a). So when reader black and white, they can see the contrast.
27. Line 289: table 5 and table 6 are to be switched. Please check.
28. Line 285: what is the correlation, need to explain the sentence further.
29. Line 290: Change the thickness or color contrast of the two lines in the graph.
30. Line 295: figure 11 does not contain any arrows.
31. Line 304-305: caption is not clear. Please revise to better caption arrangement.
32. Line 306: suggest to put the xrd result of the coating separately at this subsection.
33. Line 316: table 5 change to table 6.
34. Line 317: how were 0.1nm and 0.02nm obtained?
35. Line 322: cannot find the table that shows the microhardness measurement of powders and respective coatings.
36. Line 362: does ca coating have lower hardness? No table on the hardness of the coating was shown. Need to give the description on why CA coating has lower hardness when the CA powder is harder.
37. Line 375: Need to state what is the documented effect of powder size on DE. Should not just mentioned it is well documented without further telling the effect.
All the best, thank you.
Author Response
Hello,
Thank you for your responses in the manuscript. We will make the necessary cosmetic changes and grammatical/typing mistakes will be rectified in the new revision.
With regards to technical comments we have tried to address them as best as we can.
Please see the attachment.

Reviewer 2 Report
Dear Authors,
please see the attached pdf file

Author Response
Hello,
Thank you for reviewing the manuscript. Please see attached with the responses for your comments. These corrections will be made in the revision.

Reviewer 3 Report
The results shown in this study are interesting. However, once the manuscript was reviewed, this reviewer found some points that should be clarified by the authors.
Ø In “The deposition efficiency of the process was simply calculated as the ratio of the coating weight vs. the powder amount fed for the respective cycle time excluding technological losses.”. Be precise, indicate what you mean by technological losses.
Ø In the micrographs of figure 2 it is difficult to observe the value of the scales.
Ø Discussion of Figure 2d was omitted in section “Powder Characterization”.
Ø In “Table 5 represents the micro and nano hardness measurements on both powders.”, is there an error?, in Table 5 it says “Table 5 Crystallite size of both powders and coatings calculated from the XRD data”. Please clarify or correct.
Ø In figure 11 they must include the scale bar for each micrograph. The caption is confusing, it is not understood. Delete “at xxx mm” and indicate the magnification, also indicate which coating each micrograph corresponds to.
Ø In “The arrow in the figures indicates the 295 direction of particle travel during deposition”, in figure 11 the arrow is not found.
Ø In the Materials and Methods section it is indicated that corrosion tests were carried out, however the results were not presented. Please clarify this.
Author Response
Hello,
Thank you for reviewing the manuscript. Please see attached with the responses for your comments. These corrections will be made in the revision,

Round 2
Reviewer 3 Report
I appreciate your attention to my comments.